# Filamentous fungus-produced human monoclonal antibody provides protection against SARS-CoV-2 in hamster and non-human primate models

Franziska K. Kaiser [1,11], Mariana Gonzalez Hernandez [1,11], Nadine Krüger [2,11], Ellinor Englund[3,11], Wenjuan Du[4], Anna Z. Mykytyn [5], Mathijs P. Raadsen [5], Mart M. Lamers[5], Francine Rodrigues Ianiski [6], Tatiana M. Shamorkina [6], Joost Snijder [6], Federico Armando[7], Georg Beythien [7], Malgorzata Ciurkiewicz [7], Tom Schreiner [7], Eva Gruber-Dujardin[2], Martina Bleyer[2], Olga Batura[2], Lena Erffmeier[2], Rabea Hinkel [2], Cheila Rocha[2], Monica Mirolo[1], Dubravka Drabek [8], Berend-Jan Bosch[4], Mark Emalfarb [9], Noelia Valbuena[9], Ronen Tchelet[9], Wolfgang Baumgärtner [7], Markku Saloheimo[3], Stefan Pöhlmann [2], Frank Grosveld [8], Bart L. Haagmans [5,11] ✉ & Albert D.M.E. Osterhaus [1,10,11] ✉

Monoclonal antibodies are an increasingly important tool for prophylaxis and treatment of acute virus infections like SARS-CoV-2 infection. However, their use is often restricted due to the time required for development, variable yields and high production costs, as well as the need for adaptation to newly emerging virus variants. Here we use the genetically modified filamentous fungus expression system *Thermothelomyces heterothallica* (C1), which has a naturally high biosynthesis capacity for secretory enzymes and other proteins, to produce a human monoclonal IgG1 antibody (HuMab 87G7) that neutralises the SARS-CoV-2 variants of concern (VOCs) Alpha, Beta, Gamma, Delta, and Omicron. Both the mammalian cell and C1 produced HuMab 87G7 broadly neutralise SARS-CoV-2 VOCs in vitro and also provide protection against VOC Omicron in hamsters. The C1 produced HuMab 87G7 is also able to protect against the Delta VOC in non-human primates. In summary, these findings show that the C1 expression system is a promising technology platform for the development of HuMabs in preventive and therapeutic medicine.

In recent years, industrially produced immunoglobulins have become an important biopharmaceutical tool not only for the treatment of cancer, autoimmune and inflammatory diseases[1–3], but also as therapeutics for the prevention and treatment of bacterial and viral infections, such as respiratory syncytial virus and Ebola virus[4,5]. The COVID-19 pandemic has highlighted the importance of such prophylactics, especially the early post-infection use of SARS-CoV-2 neutralizing human monoclonal antibodies (HuMabs) in individuals with high risk of hospitalization and development of severe COVID-19[6,7]. Similarly, therapeutic treatment of COVID-19 patients with HuMabs proved to be effective in reducing fatal outcomes[8,9]. Although the health benefits of HuMabs necessitates large-scale production and

**Fig. 1 | Biochemical characterization of C1-produced HuMab 87G7.** Stained SDS gel (**A**) and Western (**B**) analysis of the fermentation culture supernatants of the C1 strain expressing HuMab 87G7. Lane 1, HEK293T cell-produced 87G7, lanes 2–5, supernatant samples from days 3–6 of the culture. Reduced and non-reduced samples are shown as indicated above the panels. The antibody used for HC detection was Anti-human IgG F(c) Goat Polyclonal Antibody (IRDye700DX) and the one used for LC was Goat anti-Human Kappa Light Chain Secondary Antibody (DyLight 800). **C** C1-produced HuMab 87G7 after purification in stained SDS gel analysis. Lane 1, reduced (R) C1-produced HuMab 87G7; lane 2, non-reduced (N-R) C1-produced HuMab 87G7; lane 3, non-reduced (N-R) HEK293T cell-produced HuMab 87G7. The experiment was performed twice, data from a representative experiment are shown.

usage, the actual usage in high-risk patients is limited by costly and complicated production procedures, even in high-income countries. An additional factor preventing more effective use of HuMabs during the COVID-19 pandemic was the continually evolving capacity of emerging variants of concern (VOCs) to escape neutralizing antibodies present in previously infected or vaccinated individuals. Therefore, there is an ongoing need to more rapidly generate HuMabs that are effective against newly emerging VOCs[10–12], given that antigenic evolution of the SARS-CoV-2 spike protein may outpace the time-consuming development process, production, testing, and approval by regulatory authorities.

The critical time and cost drivers underlying commercial HuMab development and production are mainly connected to the mammalian protein expression systems which are most commonly used for such activities[13]. To overcome these limitations we explored an alternative process of recombinant protein production in the filamentous fungus *Thermothelomyces heterothallica* (C1). This expression system has a naturally high biosynthesis capacity for secretory, biomass-hydrolyzing enzymes, which has been further enhanced by genetic engineering of the wild-type fungus to achieve high production yields of industrially used enzymes[14–16]. In recent decades, the C1 expression system has been continually genetically modified resulting in a well-characterized genetic toolbox and technology platform[17]. In addition to industrial applications, the C1 expression system has been used to express heterologous viral proteins such as the full S protein or receptor-binding domain (RBD) of SARS-CoV-2 and the hemagglutinin and neuraminidase glycoproteins of influenza virus, that both induce protective immunity against virus challenge in the mouse model[18–20].

In this work, we develop a C1 expression system for the well-characterized HuMab 87G7. This antibody is derived from H2L2 transgenic mice encoding the human immunoglobulin variable region immunized with the SARS-CoV-2 S protein. It binds to a patch of hydrophobic residues within the RBD of the SARS-CoV-2 Spike (S) protein and has broadly neutralizing activity against the VOCs Alpha, Beta, Gamma, Delta, and Omicron (BA.1/BA.2)[21]. We characterize the in vitro activity of C1-derived HuMab 87G7 and demonstrate efficacy for

prophylactic and therapeutic use in hamsters and non-human primates, in the absence of antibody-mediated enhanced virus replication.

## Results

### Antibody expression and purification
To produce HuMab 87G7 in the C1 expression system, codon-optimized genes encoding the heavy chain (HC) and light chain (LC) of HuMab 87G7 (IgG1 isotype) were synthesized. The native cellobiohydrolase 1 (CBH1) signal sequence was added to the N-terminus of both chains. The newly generated 87G7 HC and LC expression vectors were transformed into the C1 strain DNL155 which has a deletion of 14 protease genes. Upon transformation, the two vectors undergo recombination into the *cbh1* target locus. Transformants were screened using 24-well cultures followed by Western blot analysis, and the best mAb producers were purified through single colony cultures.

C1-expressed HuMab 87G7 preparations were produced by performing fermentation cultures in fed-batch process followed by antibody purification with protein A affinity chromatography. Two HuMab 87G7 preparations were produced, for use in hamster and NHP in vivo challenge studies. The purification yield of HuMab 87G7 with protein A affinity column was up to 1.6 g/L. Heavy and light chains in the C1-produced HuMab 87G7 and the HuMab 87G7 produced in HEK293T cells were identified by western blot analysis (Fig. 1A–C).

### Glycosylation pattern
The Fc effector function of IgG is modulated by specific modifications of a conserved N-linked glycan (i.e., core fucosylation) in the Fc domain. Given that N-linked glycosylation patterns of recombinantly produced biotherapeutics in C1 have potentially not been described previously, we performed glycoproteomic experiments to compare C1 and HEK293T cell-produced HuMab 87G7, and CHO cell-produced Palivizumab (Synagis, anti-RSV IgG1 HuMab). The purified recombinant antibodies were subjected to trypsin-GluC digestion, followed by reversed-phase liquid chromatography coupled to tandem mass spectrometry (LC–MS/MS). While both Palivizumab and 87G7

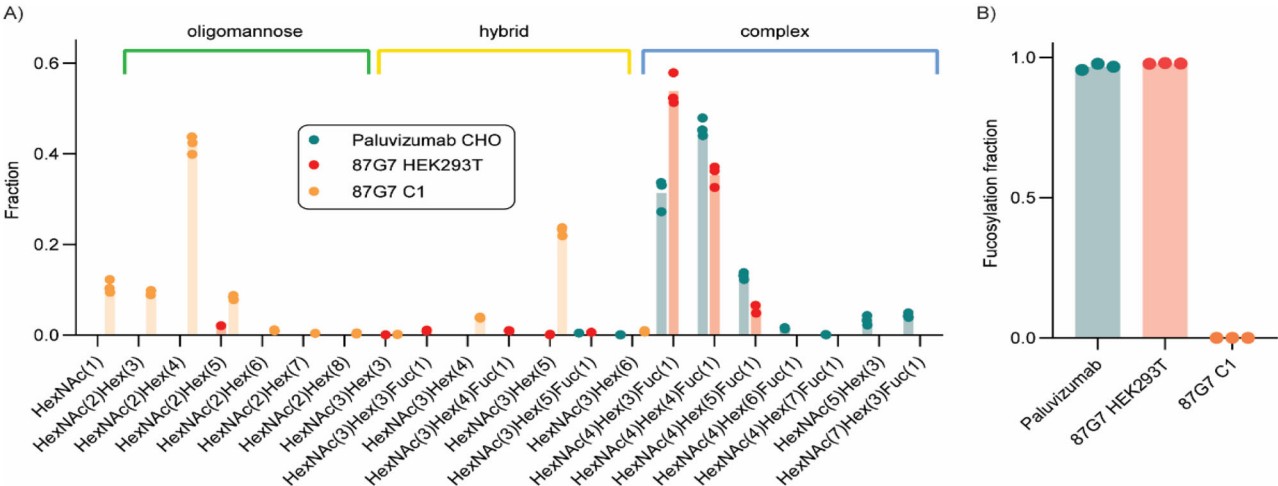

**Fig. 2 | N-linked glycosylation pattern of C1-produced HuMab 87G7.** Purified recombinant antibody was digested with trypsin-GluC and analyzed by LC–MS/MS. **A** Shown are the integrated peak areas for the specified glycoforms based on triplicate analyses. **B** The sum of all glycoforms containing a fucosylation ($n = 3$).

produced from CHO and HEK293T cells respectively, contained predominantly complex biantennary glycans of similar compositions, only oligomannose and hybrid glycans were observed in HuMab 87G7 produced in C1 (Fig. 2). The N-linked glycans present on Palivizumab produced from CHO cells and HuMab 87G7 from HEK293T cells were >95% fucosylated, while this modification was completely absent from HuMab 87G7 produced in C1. The MS/MS spectra of fucosylated N-glycopeptides from HuMab 87G7 produced in HEK293T cells contain prominent diagnostic fragment ions supporting core fucosylation (Supplementary Fig. S1).

### In vitro characterization
The fungal and mammalian HEK239T cell-derived HuMab 87G7 had comparable SARS-CoV-2 binding and neutralization properties. C1- and HEK293T cell-produced 87G7 bound to the SARS-CoV-2 S RBD and ectodomain trimer with similar efficiencies as demonstrated by ELISA (Fig. 3a, b). Biolayer interferometry (BLI) analysis showed that C1-produced HuMab 87G7 strongly binds to the trimeric S ectodomain, with comparable picomolar binding affinities as for HEK293T cell-produced 87G7 (Fig. 3c, d). Consistent with the binding data, C1- and HEK293T cell-produced HuMab 87G7 showed similar neutralization potency of SARS-CoV-2 S pseudotyped virus (Fig. 3e) and live wt (614G) SARS-CoV-2 (Fig. 3f) as well as different VOCs (Alpha, Beta, Gamma, Delta and Omicron)(Supplementary Fig. S2). In contrast, C1-produced HuMab 87G7 activated NK cells more efficiently compared to HEK293T cell-produced HuMab 87G7 (Fig. 4a), whereas binding of both antibodies to the S protein was similar (Fig. 4b). Activation of the NK92.05-CD16 cells was revealed by staining for the human degranulation marker CD107a and subsequent FACS analysis.

### In vivo protection in hamsters
The protective efficacy of C1-produced HuMab 87G7 was first assessed in a Syrian hamster challenge experiment using the SARS-CoV-2 Omicron BA.1 variant. A prophylactic strategy was used in which the C1 and the HEK293T cell-produced HuMab 87G7 as well as control HuMab Palivizumab were administered 24 h before infection by intraperitoneal injection at a concentration of 20 mg/kg. The therapeutic activity of the C1-derived antibody was evaluated by administration 12 h after infection with $10^4$ TCID$_{50}$ of Omicron BA.1 variant. Viral titers were reduced by 1 log in the nasal turbinates, with no virus detected in the lungs of animals in which antibody had been administered 24 h before infection (Fig. 5a, b). Administration of the C1-derived HuMab 87G7 12 h after infection resulted in only a minor reduction in viral titers in both lung and nasal turbinates (<1 log reduction) (Fig. 5a, b). A

reduction of viral antigen expression in the lungs as determined by immunohistochemistry using an antibody against the nucleocapsid protein, was observed in the prophylactic group, but not in the therapeutic group, in comparison to control tissues (Fig. 5c). Similarly, in the nasal cavity, scores for viral antigen expression were significantly lower for both treatment groups compared to control animals (Fig. 5d). Although we did not include a HEK293 therapy control group, earlier studies using the same antibody already showed a beneficial effect of the antibody when given after inoculation of the animals with the SARS-CoV-2 614G virus[21].

Histopathological analysis showed a protective effect of the HuMab 87G7 treatment compared to Palivizumab-treated control groups in nasal turbinate and lung tissue. A significant reduction in histopathological lesions was observed in the respiratory ($P < 0.01$) and olfactory epithelium ($P < 0.01$) of the nasal turbinates, and in sections of whole lung lobes ($P < 0.05$) of animals treated therapeutically with C1-derived HuMab 87G7 (Fig. 6a–d and Supplementary Figs. 4–8). Within the nasal cavity, the protective effect was more prominent in the therapeutically treated animals, while lung lesion scores were lower in the prophylactic group. These results show that this antibody confers protection against SARS-CoV-2 infection-related pathology when administered prophylactically, independent of the production method.

### In vivo efficacy in non-human primates
The in vivo protection afforded by C1-produced HuMab 87G7 against SARS-CoV-2 infection was also assessed in non-human primates. Rhesus macaques were administered intravenously C1-produced HuMab 87G7 prophylactically (25 mg/kg) or therapeutically, the latter in a high (25 mg/kg) or low dose (2.5 mg/kg). Palivizumab was given prophylactically as a control HuMab (25 mg/kg). Due to the milder disease observed in this species following experimental infection with VOC Omicron compared to Delta[22], animals were inoculated with $1 \times 10^5$ PFU of SARS-CoV-2 Delta (B.1.617.2) and necropsies were performed 4 days post-infection. Prophylactic treatment with the antibody provided complete protection, no viral RNA was detected in pharyngeal swabs on day 2, day 4 or in tissue samples of the lower and upper respiratory tract (Fig. 7a–h). This data is supported by virus isolation data from the same sample sites (Supplementary Table 2). No significant difference in viral RNA detection was observed between the therapy and control groups.

In the lung, all prophylactic and treatment groups exhibited lower histopathological lesions scores compared to the control group (Fig. 8a and Supplementary Fig. 9). The difference was most prominent

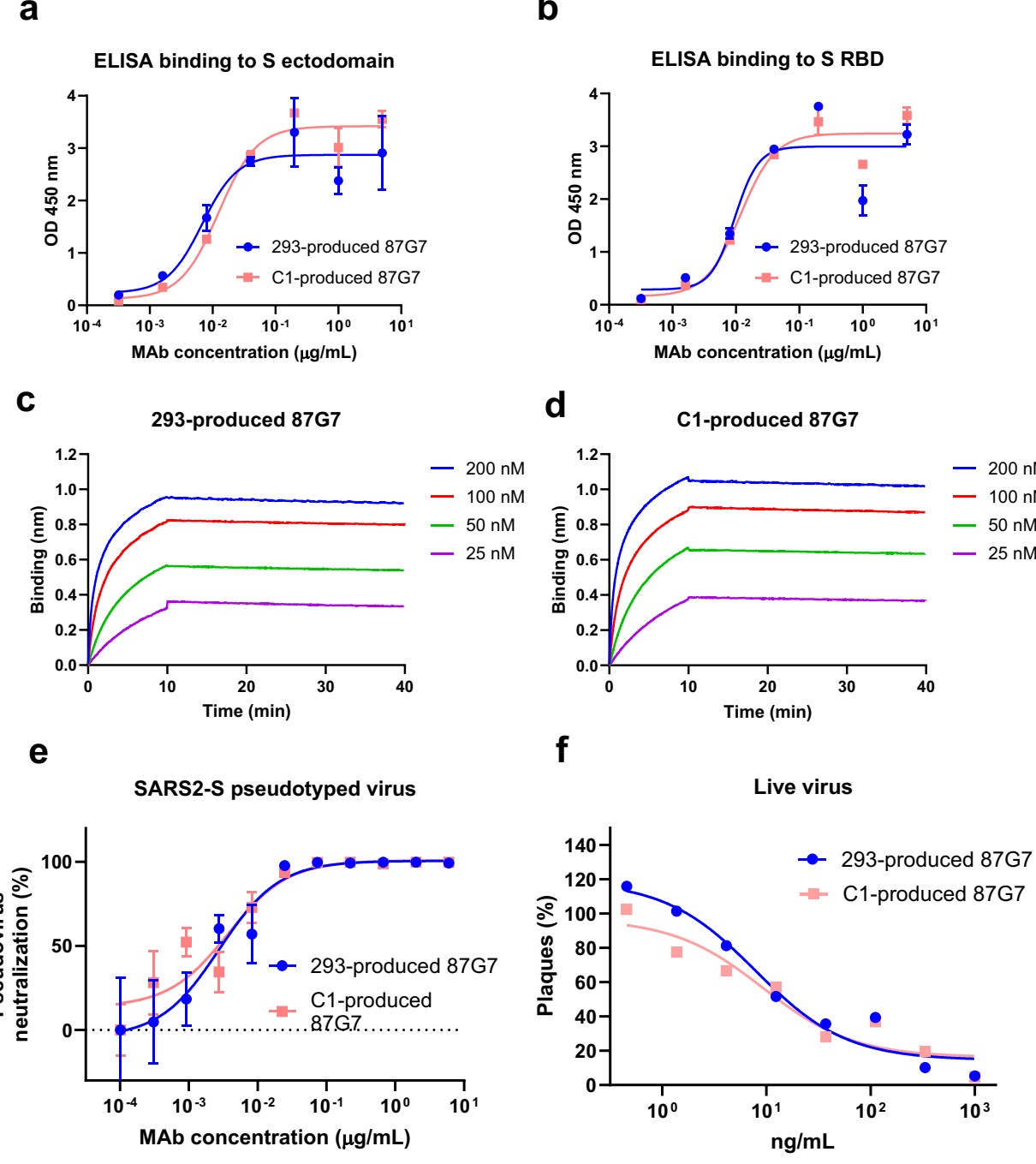

**Fig. 3 | C1 and HEK293T cell-produced HuMab 87G7 bind and neutralize SARS-CoV-2 with comparable efficiency. a, b** ELISA binding curves of C1 and HEK293T cell-produced 87G7 to the plate-immobilized RBD or ectodomain of SARS-CoV-2 spike (S). Error bars indicate SDM between three independent replicates. **c, d** Binding kinetics of C1 and HEK293T cell-produced human monoclonal antibody (Mab) 87G7 to SARS-CoV-2 S measured by biolayer interferometry (BLI). The experiment was performed twice, data from a representative experiment are shown. **e, f** Neutralizing activity of HuMab 87G7 against vesicular stomatitis virus (VSV) particles pseudotyped with ancestral SARS-CoV-2 S (Wuhan-Hu-1 strain). Inhibitory Concentrations 50% (IC50) of HuMab 87G7 against VSV-pseudotyped with SARS-CoV-2 S are displayed (Error bars indicate SEM, $n = 6$). **f** Neutralizing activity of HuMab 87G7 against SARS-CoV-2 S (614G). The experiment was performed twice, data from a representative experiment are shown ($n = 2$). Source data are provided as a Source Data file.

and statistically significant for the prophylactic group. Semi-quantitative assessment of histopathological lesions in the nasal turbinates, trachea, and stem bronchi showed lower scores in the treatment groups compared to the control group (Fig. 8b, c and Supplementary Figs. 10 and 11). Notably, the prophylactic group displayed the most prominent protective effect in the nasal turbinates. However, no statistically significant differences were detected among the groups regarding the total lesion score in these organs. In all groups and organs no SARS-CoV-2 antigen could be detected by Immunohistochemistry.

## Discussion

The prophylactic and therapeutic use of HuMabs against infections with respiratory viruses such as SARS-CoV-2, is largely limited by long

**a** 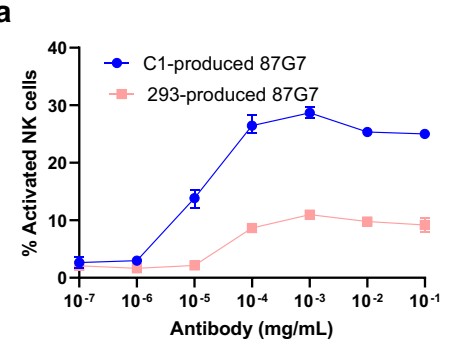  **b** 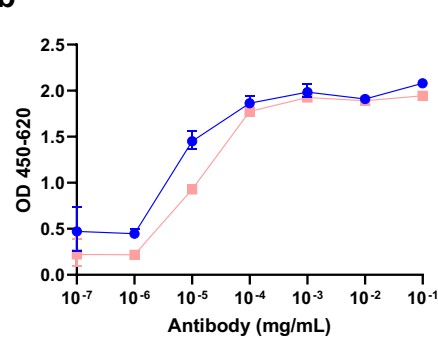

**Fig. 4 | C1 and HEK293T cell-produced HuMab 87G7 differently activate natural killer (NK) cells. a** NK cell activation curves of C1 and HEK293T-produced 87G7 bound to the plate-immobilized ectodomain of SARS-CoV-2 S. **b** Binding of C1 and HEK293T-produced HuMab 87G7 to the plate-immobilized ectodomain of SARS- CoV-2 S as measured by ELISA using the same plates that were used in (**a**), to serve as an internal control to verify similar spike binding characteristics of the antibodies. Error bars indicate SDM between three independent replicates. Source data are provided as a Source Data file.

development times, high development and production costs, and a need for continuing replacement to ensure effectiveness against newly emerging VOCs. To overcome these challenges, we investigated an alternative production system based on a genetically engineered *Thermothelomyces heterothallica* C1 strain DNL155, to express the previously described neutralizing HuMab 87G7 directed against the SARS-CoV-2 RBD[21]. The resulting C1-produced HuMab was extensively analyzed for in vitro characteristics and for in vivo efficacy in hamster and non-human primate models for SARS-CoV-2 infection. The development of an alternative expression system has the potential to overcome some of the challenges associated with traditional HuMab production, thereby providing a promising alternative for the development of effective treatments against infectious and non-infectious diseases.

Xenogeneic expression in genetically engineered C1 fungus provides several advantages compared to mammalian expression systems. *Thermothelomyces heterothallica* (C1) is known for its high biosynthesis activity of secretory enzymes with engineered modifications including several knock-out mutations to inhibit secretion of intrinsic molecules such as proteases[17]. Production cycles are much shorter compared to common manufacturing systems, thus enabling rapid development and adaptation to evolving virus variants. This could enable large-scale production of HuMabs at higher yields, in smaller, less complex production plants with simpler fermentation media compositions, making the process less costly. C1 has been developed to express and produce a variety of other proteins for therapeutic treatments, mainly mAbs but also proteins defined as "difficult to express" such as bi-specific and tri-specific antibodies and Fc-fusion proteins. This technology could be used to express other mAbs (including antibodies against Rift Valley Fever virus and ZIKA virus) to high levels in 7-day fermentation. However, a direct comparison of yield between the C1 system and the transient HEK293 system or other expression systems is difficult to make as we have not optimized the different expression systems using the same construct. Ideally, a comparison would be needed with stably transduced CHO cells, which is the generally used strategies for mAb production. C1 fermentation demonstrates significant benefits over CHO production as the C1 fermentation is based on fed-batch technology for 4–7 days with glucose feeding and with a defined low-cost medium and a wide range of conditions can be applied (pH: 5–8, Temp: 20 °C–45 °C). In addition, the current strain doesn't sporulate and the low-viscosity culture allows relatively low power input compared to other fungal cultures. The protein production requires no inducer and the protein is typically secreted to the medium. The fermentation process can be easily scaled up (the largest fermentation volume with C1 so far was 500 m³). Overall, the timeline for development of stable

cell lines and production of purified protein through protein A purification is ~6 weeks after gene synthesis, and the timeline from freezer to end of fermentation is much shorter. Yeasts on the other hand, including S. cerevisiae, Pichia pastoris, Yarrowia lipolytica, and Hansenula polymorpha, are among the current heterologous expression platforms alternatives to CHO systems which are known to be robust, easy to genetically manipulate, cost-effective, and unlike E. coli possess native PTM machinery and lack endotoxins[23]. However, in few cases, the expression level reached over 1–10 g/L[24]. Under normal conditions, the protein production is obviously lower, especially with the expression of complex proteins. In addition, P. pastoris cannot produce or secrete all proteins to such titers. In comparison, C1 yields are generally higher, although this has not been studied head-to-head. C1 can use glucose and other cheap complex carbon source for fermentation while P. pastoris process has traditionally methanol and glycerol as the carbon sources. C1 can be fermented at variable conditions (pH: 5–8, Temp: 20 °C–45 °C) in comparison to yeasts (typically 30.0 °C and pH 5.0–6.0).

In addition, several studies have demonstrated the promise of C1-produced proteins as vaccine candidates for different viral pathogens such as Influenza A virus, SARS-CoV-2, and orthobunyaviruses[18,19,25]. These studies have used different subunits, such as the hemagglutinin for Influenza A virus, the RBD for SARS-CoV-2 and part of the Gc glycoprotein for Schmallenberg virus. Toxicology testing of C1-derived SARS-CoV-2 RBD protein in rabbits did not show any local or systemic side effects of fungal-produced proteins, indicating its potential safety for use in humans[26].

Since the global spread of Omicron variants of SARS-CoV-2, most FDA-approved and commercially available HuMabs have lost their neutralization capacity[27,28]. In this study, we selected the well-characterized human IgG1 anti-SARS-CoV-2 antibody HuMab 87G7 as a candidate for production in the fungal C1 expression system[21]. Du and colleagues have previously shown that HuMab 87G7 binds to a patch of hydrophobic residues of the ACE-2 binding site resulting in a broad neutralization capacity against different VOCs (Alpha, Beta, Gamma, Delta and Omicron BA.1/BA.2). Moreover, HEK293T cell-produced HuMab 87G7 provided prophylactic protection and showed therapeutic activity in hamster and mouse models against SARS-CoV-2 challenge infections[21]. Unfortunately, the activity of HuMab 87g7 against contemporary, circulating variants, including XBB1.5, BA.4/5, and BQ.1.1. is lost, primarily due to amino acid substitution at F486[21].

In vitro characterization of binding kinetics and breadth of neutralization capacity demonstrated comparable characteristics for C1 and HEK293T cell-produced HuMabs. Binding to the RBD and S ectodomain occurs with similar efficiency for both antibody expression systems. There were differences in the N-linked glycosylation patterns

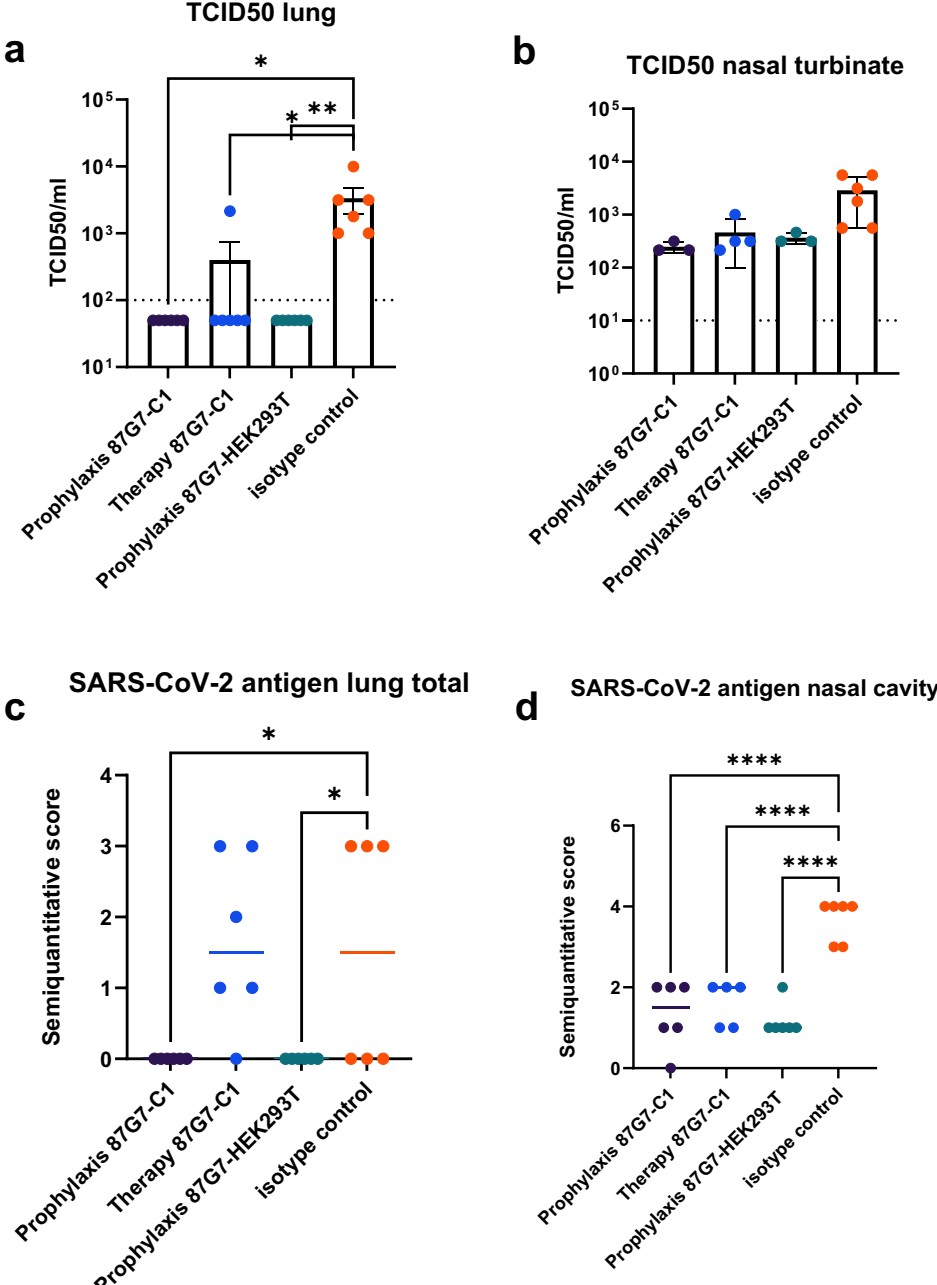

**Fig. 5 | 87G7 produced in C1 and HEK293T confers protection in hamsters.** Syrian golden hamsters were challenged with SARS-CoV-2 after or before antibody application. Lung and nasal turbinates were collected 4 days after infection. Viral titers (median) were calculated using TCID50 in **a** lung and **b** nasal turbinates. SARS-CoV-2 nucleoprotein was detected by immunohistochemistry in **c** lung and **d** nasal cavity. Examples of tissues used for quantification are provided in Supplementary Figs. 4–8. Error bars indicate SEM, $n = 6$ animals/group. One-way ANOVA was used to evaluate the statistical differences (****$P < 0.0001$; **$P < 0.01$; *$P < 0.05$). Source data are provided as a Source Data file.

between the C1-derived HuMab and the mammalian cell-expressed one. The C1-derived HuMab exhibited oligomannose and hybrid glycans while the mammalian-expressed one predominantly had complex glycans. This altered glycosylation, particularly afucosylation, could be responsible for an increased activation of NK cells observed with the C1-derived antibody compared to its mammalian-expressed counterpart. Importantly, it has been demonstrated that anti-SARS-CoV-2 monoclonal antibodies with optimized Fc domains show superior potency for prevention or treatment of COVID-19[29]. This phenomenon has also been reported for anti-PD-L1 monoclonal antibodies, Rituximab, and other monoclonal antibodies in which afucosylation

enhances binding, and boosts antibody-dependent cell cytotoxicity, thereby increasing therapeutic efficacy[30,31]. Although afucosylation and the resulting enhanced antibody-mediated cell cytotoxicity may be associated with a more severe course of COVID[32], we did not observe adverse effects of C1-derived HuMab 87G7 with respect to pathological changes seen in our in vivo studies. Additionally, effects of the fungal glycosylation patterns on long-term clinical parameters such as serum half-life requires further investigation.

The prophylactic and therapeutic activity of C1-derived HuMab 87G7 observed in the hamster model serves as a proof of principle for validating the efficacy of human monoclonal antibodies expressed in

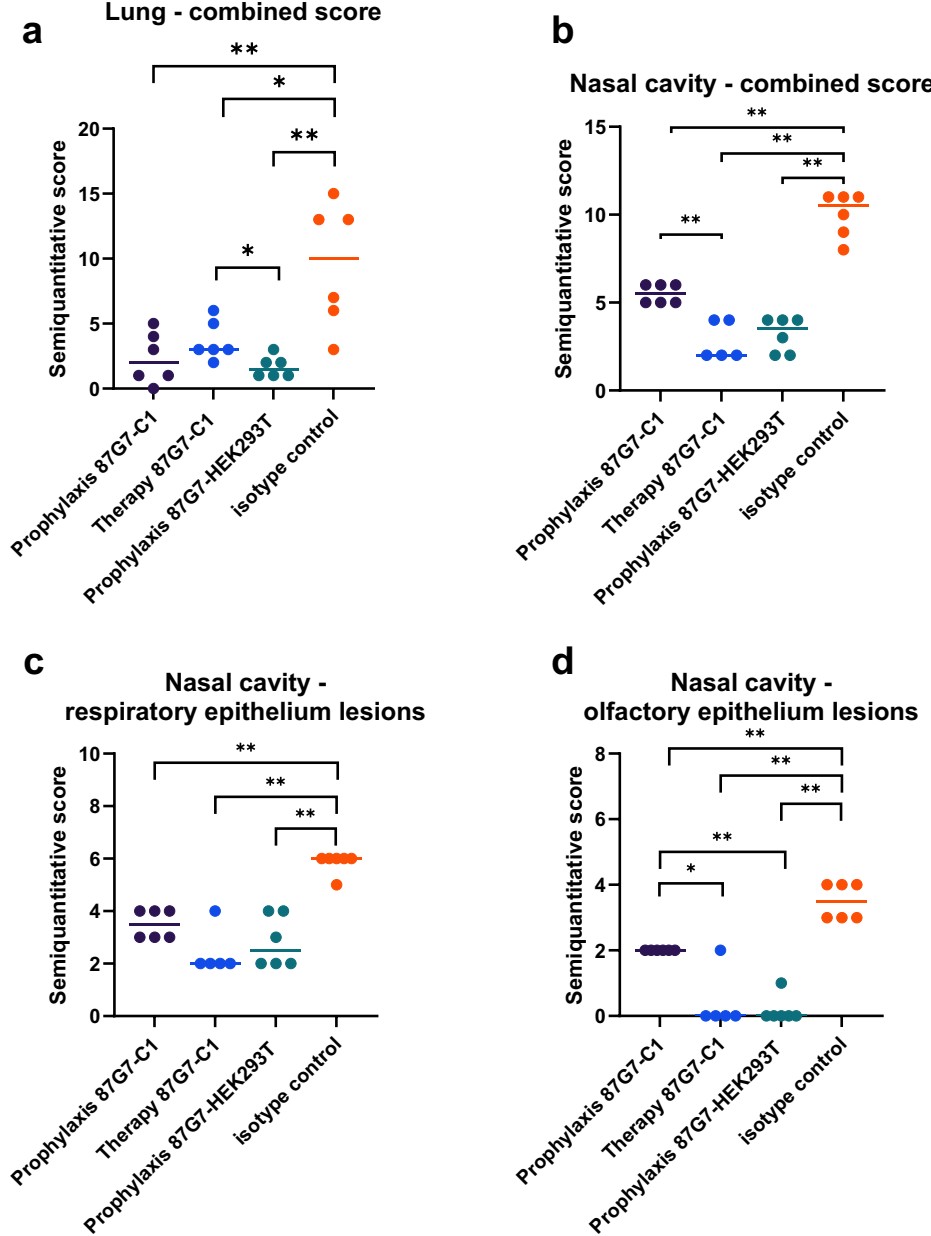

**Fig. 6 | Scoring of histopathological lesions in hamster lung and nasal cavity.** Semiquantitative scoring was performed for lung tissue (**a**) and nasal cavity in total (**b**), as well as for respiratory (**c**) and olfactory (**d**) epithelium separately. Bars indicate median values, $n = 6$ animals/group. Data were tested by two-tailed Mann−Whitney-$U$-Test to evaluate the statistical differences (**$P < 0.01$; *$P < 0.05$). Source data are provided as a Source Data file.

the C1 fungal expression system. Similarly, protection was also observed in rhesus macaques upon prophylactic treatment, even though the viral infection in this model is reduced compared to the hamster model. This is consistent with other studies reporting SARS-CoV-2 infection of hamsters and macaques[33–35]. Nevertheless, the rhesus macaque model provides evidence of protection in non-human primates, which could inform the design of future clinical trials of C1-produced HuMab's. Importantly, the prophylactic and therapeutic administration of C1-produced HuMab 87G7 at high or low doses did not show any apparent adverse effects, or antibody-dependent enhancement of virus replication.

Collectively, this study offers proof of principle that HuMabs expressed in genetically engineered C1 filamentous fungus have the potential to supersede mammalian cell-produced HuMabs for the prevention and treatment acute respiratory virus infections. As was previously shown for C1-produced proteins as vaccine candidates[18,19,25], C1-produced HuMabs may probably also be expected to be efficient to manufacture at relatively low cost, with short production cycles, enabling rapid adaptations of a scalable production system. Consequently, the C1 HuMab production platform should be further evaluated as an alternative for traditional mammalian cell-based HuMab production platforms, to overcome some of the limitations of current HuMab production, thereby improving the global access to HuMabs.

## Methods

### C1 fungal antibody expression
Codon optimization and gene synthesis were performed by Genscript (Piscataway, NJ, USA). Sequences overlapping with the C1 expression vectors were added to both ends of the synthesized HC and LC fragments. These fragments were cloned under the *bgl8* promoter in two

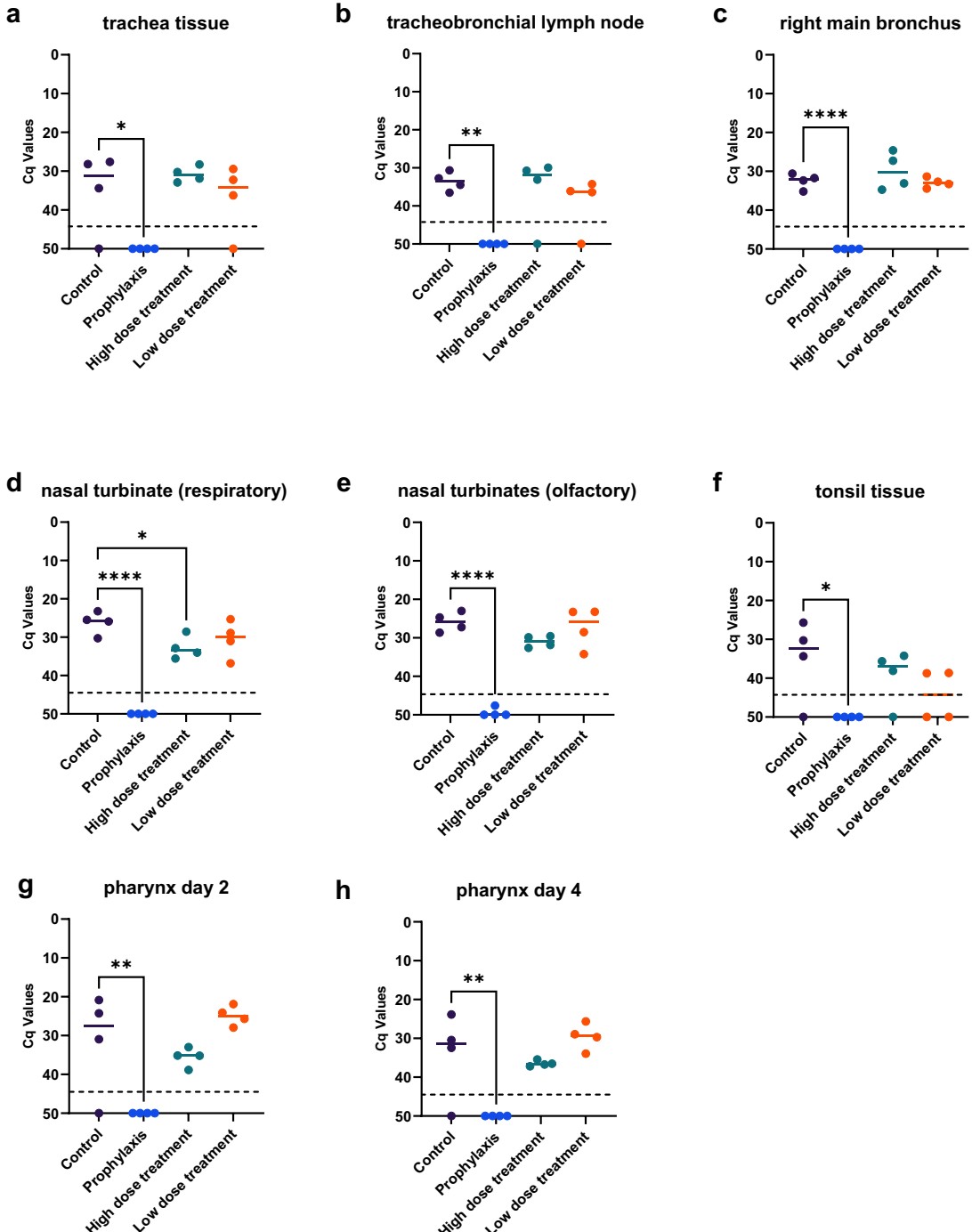

**Fig. 7 | Efficacy of C1-produced 87G7 antibody in non-human primates.** Groups of *n* = 4 macaques were treated 24 h before or 24 h after SARS-CoV-2 challenge with 25 mg/kg or 24 h after challenge with 2.5 mg/kg C1-derived 87G7. Cq values were calculated from tissue samples of trachea (**a**), tracheobronchial lymphnode (**b**), right main bronchus (**c**), respiratory (**d**), and olfactory (**e**) nasal turbinates and tonsil tissue (**f**), as well as from pharyngeal swab samples taken day 2 (**g**) and day 4 (**h**). Bars indicate median values. One-way ANOVA was used to evaluate the statistical differences (****$P < 0.0001$; **$P < 0.01$; *$P < 0.05$). Source data are provided as a Source Data file.

expression vectors carrying either the 5′ or 3′ flanking region targeting sequences for the *cbh1* (cellobiohydrolase 1) locus. Each of the vectors also has overlapping fragments of the hygromycin resistance gene. Upon transformation, the flanking region sequences undergo recombination with the *cbh1* locus and the overlapping marker fragments undergo recombination with each other. This results in integration of the HC and LC expression cassettes in opposite orientations, facing inwards, to the *cbh1* locus and conversion of the marker gene into a fully functional form. The HC gene was released from the Genscript

vector by MssI digestion and cloned with Gibson Assembly (HiFi DNA Assembly Cloning Kit, New England Biolabs, Ipswich, MA, USA) method into the PacI site of the C1 expression vector pMYT0070. The LC genes were cloned similarly into the vector pMYT0069. The fragments containing the expression cassette, overlapping fragments of the selection marker, and the 5′ and 3′ integration sequences were released from the HC expression plasmid pMYT1705 and the LC expression plasmid pMYT1704 with MssI digestion and co-transformed into the C1 production strain DNL155, where 14

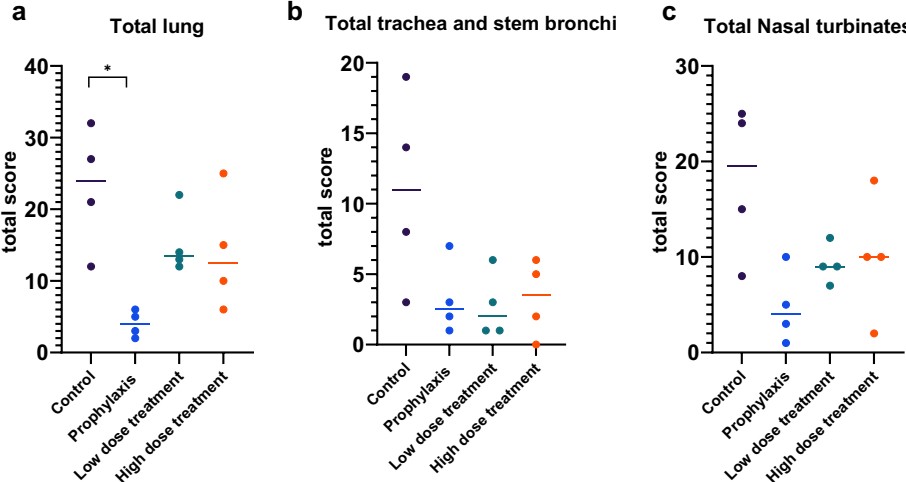

**Fig. 8 | Scoring of histopathological lesions in respiratory tract of non-human primates.** Semiquantitative scoring was performed for **a** lung tissue and **b** trachea and stem bronchi and **c** nasal turbinates. Bars indicate median values, $n = 4$ animals/group. One-way ANOVA was used to evaluate the statistical differences (*$P < 0.05$). Source data are provided as a Source Data file.

protease genes have been deleted. Transformation was done as described previously[17]. Transformants were selected for hygromycin resistance, and resulting colonies were streaked on selective medium. The transformants were screened for 87G7 mAb production by growing them in 24-well plates and performing Western blot on the culture supernatants as described previously[18]. The antibody used for HC detection was anti-human IgG F(c) Goat Polyclonal Antibody (IRDye700DX, Li-Cor, Lincoln, Nebraska, USA) and the with Goat anti-human Kappa Light Chain secondary antibody (DyLight 800, Li-Cor) used for LC detection. The best producers of 87G7 were purified through single colony cultures and analyzed by PCR for correct integration into the *cbh1* locus and the absence of the *cbh1* coding region using oligonucleotide primers listed in Supplementary Table 1.

**Antibody production and purification**
Selected best production strains were cultivated in 1 L bioreactors in a fed-batch process for 6 or 7 days at 38 °C and pH 6.4 essentially as described previously[18]. 87G7 mAb was purified from the fermentation culture supernatants with MabSelect PrismA protein A resin (Cytiva, Marlborough, MA, USA) in either 1 ml, 5 ml or 30 ml column on an ÄKTA Pure automated HPLC system (GE Healthcare, Buckinghamshire, UK) according to manufacturer's protocols. The 87G7 purification yield was based on measuring the absorbance at 280 nm with a NanoDrop 2000 Spectrophotometer (Thermo Scientific, Waltham, MA, USA). The first HuMab 87G7 preparation (MT536-6) was used in the hamster studies. The purification yield from this culture was 1.1 g/l culture supernatant. A second and larger 87G7 preparation was made by culturing the final single colony purified and PCR-verified production strain M5647 derived from another transformant. This preparation denoted MT567-5, was used in the NHP studies. The protein A affinity purification yield was 1.6 g/l culture supernatant.

SDS gel electrophoresis was done in 26-well Criterion TGX Precast 4–20% Gels (BioRad, Hercules, CA, USA) stained with Page Blue staining solution (Thermo Scientific, Waltham, MA, USA) according to the manufacturer's instructions. Non-reduced samples were made by omitting β-mercaptoethanol from the sample buffer. For injection purposes the final antibody preparation was buffer exchanged under sterile conditions to PBS using PD10 desalting columns (Cytiva, 17085101) according to manufacturer's instruction. Total yield was 90 ml at 23 mg/ml.

**Expression and purification of recombinant proteins**
pCAGGS plasmids expressing the SARS-CoV-2 (Wuhan-Hu-1 strain; Genbank: NC_045512.2) 2P-stabilized S ectodomain (S protein residues 1–1,213, with a C-terminal T4 foldon trimerization motif followed by an octa-histidine tag and a Twin-Strep-tag) and the receptor-binding domain (RBD, residues 329–538, C-terminally tagged with Strep-tag) have been described before[36]. Proteins were expressed transiently in HEK293T (ATCC CRL-11268) cells and secreted proteins were purified from culture supernatants using streptactin beads (IBA) following the manufacturer's protocol.

Recombinant 87G7 antibody in human IgG1 format were expressed in HEK293T cells following transient transfection using polyethylenimine with pairs of the IgG1 heavy and light chain expression plasmids, as described previously[36]. At 18 h after transfection, the transfection mixture was replaced by 293 SFM II expression medium (Invitrogen), supplemented with sodium bicarbonate (3.7 g/l), glucose (2.0 g/l), Primatone RL-UF (3.0 g/l), penicillin (100 IU/ml), and streptomycin (100 IU/ml), GlutaMAX and 1.5% DMSO. Tissue culture supernatants were harvested 5–6 days after transfection and recombinant antibodies were purified using Protein A Sepharose (IBA) according to the manufacturer's instructions.

**Sample preparation for glycoproteomics**
Ten μg of each antibody (HEK293T, CHO, and C1 cell line produced) sample were incubated in 100 mM Tris pH 8.5, 1% sodium deoxycholate, 10 mM tris(2- carboxyethyl)phosphine, and 40 mM iodoacetamide at 95 °C for ten minutes followed by incubation in the darkness at room temperature for 30 min. Denatured, reduced and alkylated antibodies were then diluted into fresh 50 mM ammonium bicarbonate and incubated overnight at 37 °C with 0.2 μg of trypsin (Promega) and 0.1 μg GluC (Roche). Next, formic acid was added to a final concentration of 2%, and the samples were centrifuged at max speed for 20 min at 4 °C to precipitate the sodium deoxycholate and collect the peptides from the supernatants. HEK293F, CHO, and C1 cell line-produced antibody digests were then desalted using 30 μm Oasis HLB 96-well plate (Waters). The Oasis HLB resin was activated with 100% acetonitrile and subsequently equilibrated with 0.1% formic acid in water. Next, peptides were bound to the resin, washed twice with 0.1% formic acid in water, and eluted with 100 μL of 50% acetonitrile/0.1%formic acid (v/v). The eluted peptides were

vacuum-dried and resuspended in 50 μL of 2% formic acid in water.

## Mass spectrometry

Four μl of resuspended peptides were measured on an Orbitrap Fusion Tribrid (ThermoFisher Scientific, San-Jose) mass spectrometer and Orbitrap Exploris 480 (ThermoFisher Scientific, Bremen) coupled online to nanospray UHPLC system Ultimate 3000 (ThermoFisher) in duplicates on each instrument. Peptides were loaded onto an Acclaim Pepmap 100 C18 (5 mm × 0.3 mm, 5 μm, ThermoFisher Scientific) trap column and separated with 50-cm reverse-phase analytical column (75 μm inner diameter, ReproSil-Pur C18-AQ 2.4 μm resin [Dr. Maisch GmbH]). The columns were kept at 40 °C. For the low pH reverse-phase separation, binary buffer system was used consisting of buffer A (0.1% formic acid) and buffer B (0.1% formic acid/80% acetonitrile (v/v)). Peptides measured on the Orbitrap Fusion Tribrid were separated with a 60-min gradient starting at 9% buffer B and stepwise increased to 13% in 3 min, 44% in 40 min, 99% in 3 min, 99% wash out in 5 min and re-equilibration back to 9% buffer B in 10 min. Peptides measured on the Orbitrap Exploris 480 were separated with a 90-min gradient starting at 5% buffer B and stepwise increased to 13% in 3 min, 44% in 71 min, 55% in 5 min, 99% in 1 min, 99% wash out in 4 min and re-equilibration back to 5% buffer B in 6 min. The data were acquired in data-dependent mode. Orbitrap Fusion parameters for the full scan MS spectra were as follows: an AGC target of $4 \times 10^5$ at 120,000 resolution, scan range 350–2000 $m/z$, Orbitrap maximum injection time 50 ms. The MS2 spectra of ten most intense ions (2+ to 8+) were acquired at a resolution of 60,000 with an AGC target of $5 \times 10^5$, maximum injection time 250 ms, scan range 120–4000 $m/z$, and dynamic exclusion of 16 s, precursor ion selection was at 1.6 $m/z$ and fragmentation was achieved by higher-energy collisional dissociation (HCD) at NCE 30%. Once oxonium ions corresponding to the glycan fragmentation were detected in MS2 scans, the same precursor ions were subjected to either an electron-transfer/higher-energy collision dissociation ion fragmentation (EThcD) scheme (first measurement replicate) or stepped HCD (NCE 10, 30, 45%) scheme (second measurement replicate). The supplemental higher-energy collision dissociation energy for the EThcD method was set at 27%. Exploris 480 parameters for the full scan MS spectra were as follows: an AGC target of $5 \times 10^4$ at 60,000 resolution, scan range 350–2000 $m/z$, Orbitrap maximum injection time set to auto. The MS2 spectra of twenty most intense ions (2+ to 8+) were acquired at a resolution of 30,000 with an AGC target of $5 \times 10^5$, maximum injection time set to auto, scan range was set from the first mass of 120 $m/z$, and dynamic exclusion of 14 s, precursor ion selection was at 1.4 $m/z$ and fragmentation was achieved by HCD at NCE 28%. Once oxonium ions corresponding to the glycan fragmentation were detected in MS2 scans, the same precursor ions were subjected to stepped HCD (NCE 10, 28, 45%) scheme (both measurement replicates).

## Mass spectrometry data analysis

The acquired data were analyzed with Byonic (v5.0.3). The data were searched against a custom database of recombinant HuMab 87G7 and Palivizumab VH and VL antibody region sequences. Precursor mass tolerance was set to 10 and fragment mass tolerance was set to 20 ppm. False Discovery Rate (FDR) was set to 1%. Up to six missed cleavages were permitted using C-terminal cleavage at R/K/E/D (trypsin + GluC cleavage specificity). Carbamidomethylation of cysteine was set as fixed modification, methionine and tryptophane oxidation as variable common 2, N-glycan modifications as variable rare 1, allowing up to max. 2 variable common and max. 1 variable rare modifications per one peptide. A glycan database used in the search contained 305 N-linked glycans. Glycopeptide hits reported in the Byonic result file were accepted if the Byonic score was ≥200, LogProb ≥3. Accepted glycopeptides were manually inspected for quality of fragment

assignments. The glycopeptide was considered true-positive if the appropriate b, y, c, and z fragment ions were matched in the spectrum, as well as the corresponding oxonium ions of the identified glycans. All glycopeptide identifications were merged into a single non-redundant list per sequon. Glycans were classified based on HexNAc content as high-mannose (2 HexNAc), hybrid (3 HexNAc) or complex (>3 HexNAc). Byonic search results were exported to mzIdentML format to build a spectral library in Skyline (22.2.0.351) and extract peak areas for individual glycoforms from MS1 scans. The full database of variable N-linked glycan modifications from Byonic was manually added to the Skyline project file in XML format. Glycopeptide identifications from Byonic were manually inspected in Skyline and evaluated for correct isotope assignments and well-defined elution profiles, suitable for peak integration. In the case of multiple missed cleavages, reporting on the same site-specific glycoform, peak areas were summed in the semiquantitative analysis. Reported peak areas were pooled based on the number of HexNAc and Fuc to distinguish high-mannose/hybrid/complex glycosylation and fucosylation respectively. The quantified data was represented with GraphPad Prism 9.3.1 software.

## ADCC assay

Effector functions of monoclonal 87G7 produced in human cell line 293 (MC-H) and from the C1 filamentous fungal culture (MC-C1) were compared in a Natural Killer cell-based Antibody-dependent Cellular Cytotoxicity (ADCC) assay. CD16 overexpressing NK92 cells (NK92.05, a kind gift from Kerry S. Campbell) were cultured in Alpha-MEM supplemented with NaHCO3 (2.2 g/L, pH 7.2), 2-ME (0.0001 M), L-glutamine (200 mM, Gibco), myo-inositol (0.2 mM), 10% horse serum, 10% fetal bovine serum, folic acid (0.004 mM), sodium pyruvate (1 mM) penicillin (100 IU/ml), and streptomycin (100 μg/ml). NK92 cells were stimulated in RPMI 1640 supplemented with 10% FBS, 100 IU/ml, and 100 μg/ml streptomycin (R10). His-tagged, prefusion stabilized SARS-CoV-2 recombinant full spike protein (R&D system cat no. 10549) was coated on a high-binding 96 wells ELISA plate (Costar cat. No. 3590) at a concentration of 1 μg/ml in PBS O/N at 4 °C. Plates were washed twice with 200 μL PBS per well and blocked with 10% FBS in PBS for 1 h at 37 °C. After washing 3× using 200 μl of PBS with 0.05% (v/v) Tween-20 (Sigma, cat.no. P3563), 100 μL of MC-H, MC-C1, IG and HC diluted in R10 was added to the plate and incubated at 37 °C for 2 h. Monoclonals were tested in concentrations ranging from 0.1 mg/ml to 0.1 ng/ml. Serum and Nanogam in 10-fold dilution steps from 1:100 to 1:107. After washing, $10^5$ NK92 cells were added per well, suspended in 100 μl R10 supplemented with anti-CD107a-V450 (clone H4A3, 1:100), golgistop and golgiplug (0.67 μl/ml and 1 μl/ml, Becton Dickinson cat. No. 554724, 555029) and incubated for 5 h at 37 °C. NK92 cells were pelleted by centrifuging at 1300 RPM for 5 min and resuspended in R10 supplemented with golgistop and golgiplug, anti-CD56-PE (clone B159, 1:25), and aqua fluorescent reactive dye (LIVE/DEAD, ThermoFisher, cat no. L10120). After staining at 4 °C for 30 min, cells were washed in 150 μL flow cytometry buffer consisting of PBS with 10% BSA and 0.5 M EDTA (FB), supplemented by golgistop and golgiplug, fixed in 100 μl cytofix (Becton Dickinson cat. No. 554655), suspended in 150 μL FB and kept at 4 °C until analysis by flow cytometry.

After NK92 cell stimulation, the SARS-CoV-2 recombinant full spike coated plate used in the ADCC assay was washed and incubated with rabbit anti-human IgG-HRP (DAKO, cat. No. P0214, 1:6000 in PBS) for 1 h at 37 °C. After washing, plates were incubated with TMB substrate (Seracare, cat. No. KPL-507603) for 5 min at room temperature. The reaction was stopped using 0.25 M sulfuric acid (Merck cat. No. 1.09073.1000) and optical density (OD) was measured immediately using a TECAN F200 spectrophotometer at 450 nm with a 620 nm reference wavelength. NK92 cells were analyzed on a Becton Dickinson Lyric flow cytometer and FlowJo software, version 10.7.1. NK cells were

gated using forward-sideward scatter, isolation of singlets, live cells, and CD56 positive cells. Activated NK cells were identified based on CD107a expression.

## Viruses and cells

Viruses propagated for infection of hamsters and non-human primates were grown to passage 3 on Calu-3 cells, harvested 48–72 h post-infection, clarified for 5 min at $1000 \times g$, aliquoted, and stored at −80 °C until use. Virus stocks were titrated by preparing 10-fold serial dilutions in Opti-MEM I (1×) + GlutaMAX (Gibco). One hundred microliters of each dilution were added to confluent monolayers of Calu-3 cells in 12 or 24-well plates. After 4 h at 37 °C, cells were overlaid with 1.2% Avicel (FMC BioPolymer) in Opti-MEM I (1×) + GlutaMAX for 48 h. Avicel was removed, cells were fixed in formalin and permeabilized in 70% ethanol. Cells were washed with PBS and blocked in 3% bovine serum albumin (BSA; Sigma-Aldrich) in PBS. Stainings were performed using rabbit anti-nucleocapsid antibody in 0.1% BSA (Sino Biological, 1:4000) for one hour followed by goat anti-rabbit Alexa Fluor 488 (Invitrogen, 1:4000) in 0.1% BSA. Plates were washed once in PBS and scanned on the Amersham Typhoon Biomolecular Imager (channel Cy2; resolution 10 μm; GE Healthcare). Titers were determined using ImageQuant TL software. B.1.17, B.1.351, and P.1 passage 3 stocks were sequenced and without mutations in Spike compared with the original respiratory specimen. The Beta variant passage 3 sequence contained two mutations compared with the original respiratory specimen: one synonymous mutation C13860T (Wuhan-Hu-1 position) in ORF1ab and an L71P change in the E gene (T26456C, Wuhan-Hu-1 position). The Delta (B.1.617.2) and Omicron BA.1 (B.1.1.529 BA.1) variant passage 3 sequences were identical to the original respiratory specimens. All work with infectious SARS-CoV-2 was performed in a Class II Biosafety Cabinet under BSL3 conditions at Erasmus Medical Center.

HEK293T and VeroE6 cells were maintained in DMEM supplemented with 10% FCS, sodium pyruvate (1 mM, Gibco), non-essential amino acids (1×, Lonza), penicillin (100 IU/ml), and streptomycin (100 IU/ml) at 37 °C in a humidified CO2 incubator. Cell lines tested negative for mycoplasma. Calu-3 cells were maintained in Opti-MEM I (1×) + GlutaMAX with 10% FCS and penicillin and streptomycin.

## ELISA analysis of antibody binding to SARS-CoV-2 S antigens

Purified S antigens (1 μg/ml) were coated onto 96-well NUNC Maxisorp plates (Thermo Scientific) at 4 °C overnight followed by three washing steps with Phosphate Saline Buffer (PBS) containing 0.05% Tween-20. Plates were blocked with 3% bovine serum albumin (BSA, Fitzgerald) in PBS with 0.1% Tween-20 at room temperature (RT) for 3 h. Purified HuMab 87G7 was allowed to bind to the plates at 5-fold serial dilutions, starting at 6 μg/ml diluted in PBS containing 3% BSA and 0.1% Tween-20, at RT for 1 h. Antibody binding to the S proteins was determined using a 1:2000 diluted HRP conjugated goat anti-human IgG (ITK Southern Biotech) for 1 h at RT. HRP activity was measured at 450 nm using tetramethylbenzidine substrate (BioFX) and an ELISA plate reader (EL-808, Biotek).

## Antibody binding kinetics and affinity measurement

HuMab 87G7 (21 nM) was loaded onto Protein A biosensors (ForteBio) for 10 min. Antigen binding was performed by incubating the biosensor with 2-fold dilutions of recombinant SARS-CoV-2 S ectodomain trimer for 10 min followed by a dissociation step (30 min) to observe the decrease of the binding response. The apparent affinity constant ($K_D^{App}$) was calculated using 1:1 Langmuir binding model on Fortebio Data Analysis 7.0 software.

## Pseudovirus neutralization assay

The production of SARS-CoV-2 S pseudotyped vesicular stomatitis virus (VSV) and the neutralization assay were carried out as described previously[36]. In brief, HEK293T cells at 70–80% confluency were transfected with the pCAGGS expression vectors encoding SARS-CoV-2 S (Wuhan-Hu-1 virus Genbank: NC_045512.2) with a C-terminal cytoplasmic tail 18-residue truncation to increase cell surface expression levels. Cells were infected with VSV G pseudotyped VSVΔG bearing the firefly (*Photinus pyralis*) luciferase reporter gene at 48 h after transfection. Twenty-four hours later, the supernatant was harvested and filtered through 0.45 μm membrane. Pseudotyped S VSV was titrated on VeroE6 cells. In the virus neutralization assay, 3-fold serially diluted mAbs were pre-incubated with an equal volume of virus at RT for 1 h, and then inoculated on VeroE6 cells, and further incubated at 37 °C. After 20 h, cells were washed once with PBS and lysed with Passive lysis buffer (Promega). The expression of firefly luciferase was measured on a Berthold Centro LB 960 plate luminometer using D-luciferin as a substrate (Promega). The percentage of neutralization was calculated as the ratio of the reduction in luciferase readout in the presence of mAbs normalized to luciferase readout in the absence of mAb. The half-maximal inhibitory concentrations (IC50) were determined using 4-parameter logistic regression (GraphPad Prism v8.3.0).

## Plaque reduction neutralization assay

PRNT50 assays were performed as described previously. Briefly, monoclonal antibodies were 3-fold serially diluted in 60 μL Opti-MEM I (1X) + GlutaMAX (Gibco). Four hundred PFU (based on 8 h titrations) in 60 μL were added per well to a final volume of 120 μL and a serum dilution of 1:20 in the first well. Plates were incubated for 1 h at 37 °C. Next 100 μL of virus and serum mix was added to confluent monolayers of Calu-3. SARS-CoV-2 infected plates were incubated for 8 h at 37 °C before fixing in formalin and permeabilizing in ethanol. Plates were then washed in PBS and stained as described for virus titrations. Nuclei were stained with Hoechst for 30 min. Cells were imaged using the Opera Phenix spinning disk confocal HCS system and the number of GFP-positive/Alexa Fluor 488-positive infected cells were quantified using the Harmony software (version 4.9, PerkinElmer). The PRNT50 was calculated based on non-linear regression. All analyses were performed using GraphPad Prism 9 software.

## Design of hamster experiment

The hamster challenge experiment was performed at the Research Center for Emerging Infections and Zoonoses (RIZ), University of Veterinary Medicine, Hanover, Germany under BSL3 conditions. Approval was given by the Lower Saxony State Office for Consumer Protection and Food Safety (approval no.: 21-3755). The 8–10 weeks old Syrian male hamsters (*Mesocricetus auratus*) were purchased from Janvier Laboratories and housed in individually ventilated cages (IVCs, Tecniplast), starting 10 days prior to the experiment.

The hamster experiment was performed as previously described[21]. In brief, 20 mg/kg of HuMab 87G7 or a non-SARS-CoV-2 specific human IgG antibody against respiratory syncytial virus (Palivizumab) were applied intraperitoneally. The prophylaxis and the control group ($n = 6$ animals per group) received the antibody injection 24 h before SARS-CoV-2 challenge and the treatment group received the antibodies 12 h post-challenge. Hamsters were infected intranasally with 50 μl of virus suspension containing $10^4$ TCID$_{50}$ of SARS-CoV-2 Omicron BA.1 (B.1.1.529). During the experiments animals were monitored at least twice daily and humanely euthanized four days after infection. Antibody injection, virus challenge, and euthanasia were performed under isoflurane anesthesia to minimize distress for the hamsters.

During necropsy, the right lung lobes and nasal turbinates were collected and frozen at −80 °C for virological analyses. The left lung lobe and nasal turbinates (Supplementary Fig. 4a, b) were fixed in 10% buffered formalin (Chemie Vertrieb GmbH & Co Hannover KG, Hannover, Germany). Lungs were pre-fixed by injection of 10% buffered formalin to ensure optimal histopathological evaluation. Nasal turbinates were decalcified in decalcifier soft (6484.3, Carl Roth GmbH + Co.

KG, Karlsruhe, Germany) for 14 days after fixation prior to routine tissue processing. Samples were subsequently embedded in paraffin, cut into 2 µm thick serial sections, and stained with hematoxylin and eosin (H&E). Sections were scanned using an Olympus VS200 Digital slide scanner (Olympus Deutschland GmbH, Hamburg, Germany) and evaluated as previously described[34]. The evaluation was performed in a blinded manner, using a semiquantitative scoring system[21]. Subsequently, histopathological scoring was reviewed and confirmed by board-certified veterinary pathologists. One animal was excluded from the nasal cavity evaluation due to absence of nasal mucosa in the formalin-fixed paraffin-embedded tissue (FFPE) block. Immunohistochemistry for SARS-CoV-2 nucleoprotein was performed and evaluated as described previously[35,36].

### Design of non-human primate experiment
Sixteen 4–10 years-old male rhesus monkeys (*Macaca mulatta*) were kept at the German Primate Center complying with the regulations of the European Parliament and the Council Directive on the protection of animals used for scientific purposes (2010/63/EU). The experiment was approved by the Lower Saxony State Office for Consumer Protection and Food Safety (approval no.: 20-3601). All work with infectious SARS-CoV-2 was conducted under BSL3 conditions.

For the non-human primate in vivo study an established SARS-CoV-2 VOC Delta rhesus macaque model was deployed. Animals were randomly assigned to a control, a prophylaxis as well as two therapy groups, each consisting of four animals. For the control and prophylaxis group, 25 mg/kg of HuMab 87G7 or a non-SARS-CoV-2 specific human IgG antibody against respiratory syncytial virus (Palivizumab), respectively, were administered intravenously (i.v.) at one day prior to infection. Animals in therapy groups received 25 mg/kg or 2.5 mg/kg of HuMab 87G7 i.v. at one day post-infection. All animals were inoculated with $1 \times 10^5$ PFU of SARS-CoV-2 Delta (B.1.617.2) diluted in sterile PBS via the intranasal (0.25 ml per nostril) and intratracheal (4.5 ml) route. The animals were observed daily for clinical signs. Blood (EDTA and serum) and swab samples were taken at day 0, 2, and 4 p.i. and at the same days clinical parameters including bodyweight and body temperature were measured. After euthanasia at day 4 p.i., necropsies were performed and tissue samples were collected.

Necropsies were performed immediately after death. The nasal turbinates, tonsil, pharynx, trachea, stem bronchi, tracheobronchial lymphnode, and lung were sampled for virological and histological analyses (Supplementary Fig. 4C–E). Samples for virological analyses were collected and frozen at −80 °C until further processing. For histological analysis, samples were collected in 10% formaldehyde solution. The nasal turbinates were trimmed longitudinally by including the entire length of the nasal cavity. Additionally, two cross-sections of the nasal septum were prepared. The turbinates were decalcified in decalcifier soft (6484.3, Carl Roth GmbH + Co. KG, Karlsruhe, Germany) for 7–8 days. Tracheal and bronchial samples were decalcified for 3–4 days prior to processing. The left lung was fixed by intrabronchial instillation of formalin and trimmed. In total, 3 sections from the nasal turbinates, 4 sections from the trachea/extrapulmonary bronchi, and 13 sections from the right lung were analyzed for each animal. Sections were scanned using an Olympus VS200 Digital slide scanner (Olympus Deutschland GmbH, Hamburg, Germany). Lesions were scored with a semiquantitative scoring system as described previously[34], with minor modifications. The evaluation was performed in a blinded manner and subsequently confirmed by board-certified veterinary pathologists.

### Virus infectivity titration
Virus infectious titers were determined as described previously[34]. Tissue samples were homogenized using a TissueLyser II (Qiagen). Calu-3 cells were infected with 10-fold serial dilutions of tissue homogenate. Plates were further incubated in a humidified atmosphere at 37 °C, 5%

$CO_2$. Five days after infection, cells were fixed with 4% PFA and stained using rabbit anti-SARS-CoV-2 nucleocapsid antibody (Sino Biological, Peking, China-40588-T62; dilution: 1:1000). A goat anti-rabbit IgG conjugated with Alexa fluor 488 was used as the secondary antibody (Thermofisher, Waltham, Massachusetts, USA, A11008; dilution: 1:1000). Viral titers (TCID50/ml) were calculated using the "Spearman–Kärber method". Analysis were performed in quadruplicates.

### RT-qPCR
Viral RNA was extracted from tissue samples using QIAmp Viral extraction kit following the manufacturer's instructions. RT-qPCR was performed according to an established protocol[34,37]. Primers and probe targeting SARS-CoV-2 E gene were used following the Super Script III Platinum One-Step RT-qPCR (Invitrogen) protocol. Amplification was performed as follows: reverse transcription 55 °C 20 min, denaturation 95 °C 3 min, followed by 50× cycles of amplification at 95 °C 15 s, 58 °C 30 s, where data was acquired. Further analysis and Cq values were determined using the BioRad CFX Maestro software (BioRad).

### Reporting summary
Further information on research design is available in the Nature Portfolio Reporting Summary linked to this article.

## Data availability
The mass spectrometry proteomics data have been deposited to the ProteomeXchange Consortium via the PRIDE partner repository (https://www.ebi.ac.uk/pride/archive/projects/PXD041208). The reused full genome viral sequences have been deposited in GenBank and the accession numbers are as follows:

614G (https://www.ncbi.nlm.nih.gov/nuccore/OM304632),
Alpha (https://www.ncbi.nlm.nih.gov/nuccore/MW947280),
Beta (https://www.ncbi.nlm.nih.gov/nuccore/OM286905),
Delta (https://www.ncbi.nlm.nih.gov/nuccore/OM287123),
Gamma (https://www.ncbi.nlm.nih.gov/nuccore/OM442897)
and Omicron BA.1 (https://www.ncbi.nlm.nih.gov/nuccore/OM287553). All other data are available in the article and its Supplementary files or from the corresponding author upon request. Source data are provided with this paper.

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

## Acknowledgements

We thank Caroline Schütz, Julia Baskas, Kerstin Rohn, Kerstin Schöne, and Jana Svea Harre for excellent technical support and Martin Ludlow for English editing and suggestions. Part of the work was funded by the MANCO project, a European Union's Horizon 2020 research and innovation program (grant agreement No 101003651, D.D., B.J.B., F.G., B.L.H., A.O.). The study was also supported by the COVID-19 Research Network of the State of Lower Saxony (COFONI) with funding from the Ministry of Science and Culture of Lower Saxony, Germany (14–76403–184, F.A., M.C., W.B.). This research was funded by the Deutsche Forschungsgemeinschaft (DFG; German Research Foundation, G.B., L.H., F.K., W.B., and A.O.) -398066876/GRK 2485/1-VIPER-GRK. S.P. acknowledges funding by BMBF (01KI2006D, 01KI20328A, 01KX2021), the Ministry for Science and Culture of Lower Saxony (14-76103-184, COFONI Network, including projects 7FF22, 6FF22, 10FF22), and the German Research Foundation (DFG; PO 716/11-1, PO 716/14-1). J.S. is funded by the Dutch Research Council NWO Gravitation 2013 BOO, Institute for Chemical Immunology (ICI; 024.002.009). T.S. was supported by the Luxemburgish National Research Fund (FNR, Project Reference: 15686728). This open access publication was funded by the Deutsche Forschungsgemeinschaft (DFG, German Research Foundation)-491094227 "Open Access Publication Funding" and "the University of Veterinary Medicine Hannover, Foundation."

## Author contributions

The study was designed by F.G., B.L.H., and A.O. Production and expression of the C1 antibody was performed by E.E. and D.D. In vitro testing of the antibodies as done by W.D., A.Z.M., and M.P.R. Glycosylation analysis was done by F.R.I. and T.M.S., Viral strains were propagated and sequenced by M.M.L., Animal experiments were performed by N.K., E.G.D., M.B., O.B., L.E., R.H., C.R., Pathology evaluation was performed by F.A., G.B., M.C., T.S. and W.B. Immunolabeling was conducted and analyzed by F.A., and G.B. Virus titration and PCR were performed and analyzed by F.K.K., M.G.H., and M.M. Data analysis and interpretation were performed by F.A., G.B., F.K.K, M.G.H., M.C., W.B., B.L.H., and A.O. Figures were prepared by F.A., G.B., F.K.K., M.G.H., and B.L.H. The work was supervised by J.S., B.J.B, M.E. N.V., R.T., W.B., M.S., S.P., F.G., B.L.H., an A.O.The original draft was written by F.K.K., M.G.H., B.L.H and A.O. The manuscript was reviewed, edited, and approved by all authors. The project was supervised by B.L.H. and A.O.

## Funding

 

## Competing interests

A patent has been filed in the UK (2112933.3) on HuMab 87G7 described in this manuscript with authors F.G., B.L.H., and B.J.B. as inventors. Authors D.D. and F.G. are (part-time) employees of Harbour Biomed and hold company shares. Authors M.E., N.V., and R.T. are employees of Dyadic International, Inc. Author A.O. is CSO of CR2O, a CRO based in The Netherlands Authors E.E. and M.S. work for the company VTT Technical Research Centre of Finland, Ltd. The remaining authors declare that the research was conducted in the absence of any commercial or financial relationships that could be construed as a potential conflict of interest.

## Additional information

ns licence, unless indicated otherwise in a credit line to the material. If material is not included in the article's Creative Commons licence and your intended use is not permitted by statutory regulation or exceeds the permitted use, you will need to obtain permission directly from the copyright holder. To view a copy of this licence, visit http://creativecommons.org/licenses/by/4.0/.

© The Author(s) 2024

[1]Research Center for Emerging Infections and Zoonosis, University of Veterinary Medicine, Foundation, Hannover, Germany. [2]German Primate Center - Leibniz Institute for Primate Research, Göttingen, Germany. [3]VTT Technical Research Centre of Finland Ltd, 02150 Espoo, Finland. [4]Virology Section, Infectious Diseases and Immunology Division, Department of Biomolecular Health Sciences, Faculty of Veterinary Medicine, Utrecht University, Utrecht, the Netherlands. [5]Department of Viroscience, Erasmus Medical Center, Rotterdam, the Netherlands. [6]Biomolecular Mass Spectrometry and Proteomics, Bijvoet Center for Biomolecular Research and Utrecht Institute of Pharmaceutical Sciences, Utrecht University, Padualaan 8, 3584, CH Utrecht, The Netherlands. [7]Department of Pathology, University of Veterinary Medicine, Foundation, Hannover, Germany. [8]Department of Cell Biology, Erasmus Medical Center, Rotterdam, the Netherlands and Harbour BioMed, Rotterdam, the Netherlands. [9]Dyadic International, Inc, Jupiter, FL, USA. [10]Global Virus Network, Baltimore, MD 21201, USA. [11]These authors contributed equally: Franziska K. Kaiser, Mariana Gonzalez Hernandez, Nadine Krüger, Ellinor Englund, Bart L. Haagmans, Albert D.M.E. Osterhaus. ✉e-mail: b.haagmans@erasmusmc.nl; Albert.Osterhaus@tiho-hannover.de

