## [Peer Review File · Nature Communications]

Filamentous Fungus-Produced Human Monoclonal Antibody Provides Protection Against SARS-CoV-2 in Hamster and Non-Human Primate ModelsREVIEWER COMMENTS

Reviewer #1 (Remarks to the Author):

Kaiser et al have performed a comparison of monoclonal antibody production systems, mammalian versus a new fungal system, C1, using 87G7 as the Fab in IgG1 backbone against SARS-CoV-2 viruses. The experiments are clearly performed and well set out from in vitro characterisation to importantly hamster and non-human primate protection experiments. The lack of fucosylation of C1 produced antibodies is the main gain from the production method. The increase in Fc effector functions of the C1 and lack of fucosylation and shorter structures in the Fc as anti-inflammatory could be highlighted. This could also be extended to other viruses and diseases where Fc effector functions may be an advantage: Eg Rituximab fucosylation therapy is known to significantly impact tumor clearance (reviewed in <https://www.tandfonline.com/doi/abs/10.1517/14712598.6.11.1161>). Some clarity in the text would help for context and can be readily incorporated without further experiments needed.

Minor comments:

- How was the Fab of the mAb derived? Where did it originally come from? At the initial results this could be given for context even if described in previous publications. This is buried at line 220, but should be described briefly in the introduction.
- Which omicron variant used (BA.1 etc should be described) (line 535, BA.1 used, again buried in the text and could be briefly described earlier for context.
- Results line 106, Figure 1, what is the yield in comparison from the C1 vs HEK293 system? Input versus output, 1.6g/ml from C1 versus HEK293 system. If yield is the main advantage for reduced cost by C1 system, the yield or speed should be compared, a table would be helpful.
- Data not shown line 101, e.g. size integration, give the sequence (as per Nat Comms data availability?) Line 207, reference?
- The data in Figure 4 is weakly described and vague
 - o Figure 4A, what is the source of the NK cells? PBMC ex vivo or cell line? Basis of activation? CD107a or IFN γ secretion? In the methods. The CD16+ NK cell source is not given.
 - o What is the difference in Figure 4B and 3A, both appear to be S based ELISA assays, or is

Figure 4B based on FcγR3a binding of the 2 mAb? The methods, line 435, described a cell-based ELISA? And flow cytometry, but no supporting FACS plots. This method is vaguely described, and the text and figure legend should be expanded.

- Supp figure 2 and Figure 3F, Area under the curve analysis for the VNA data maybe a useful way to summarise the 6 strains tested in the main figure as a panel.

- Line 145/146, which figure panel does the 1 log reduction refer to?

- Figure 6, Prophylaxis (-24 hours pi) and treatment (+12 hours pi) labels may help for context (or consistency w figure 5). What was the control treatment? An isotype or PBS control?

- A HEK293 therapy group would have been a useful comparison, if not available due to difficulty in conducting hamster P3 experiments, a discussion point could be raised.

- What antigen detection is Figure 5CD referring to? In figure legend but not main text, and which approach was used. Nucleocapsid detection by histology ELISA flow? This could be described briefly more explicitly.

- Supp figure 4 to 8, do not include treatment comparisons. Either time course of infection or treatment types would be informative, currently nothing is gained from these figures other than tissue context.

- Supp figure 4, the boxed areas could be directly labelled, rostral and caudal and nasal septum is unclear.

- What is the treatment dose licensed for Palivizumab? 25mg/kg used. Hamsters are given mAb intraperitoneally, whilst macaque are intravenous, and slight difference in dose 20mg vs 25mg/kg, why the differences?

- Line 159, what are the statistical differences in lesions, mean stdev and p values- these stats could be described more throughout the main results text.

Reviewer #2 (Remarks to the Author):

The paper submitted by Kaiser et. al. characterizes a method for producing monoclonal antibodies by leveraging recombination and expression in the filamentous fungus *Thermothelomyces heterothallica* (C1 expression system). The SARS-CoV-2 monoclonal antibody 87G7 is used as proof of concept and antibody production, binding, glycosylation, viral neutralization, and efficacy in hamsters are compared between fungus-produced vs.

293 transient transfection-produced antibody. In vivo efficacy as prophylaxis and treatment of fungus-produced antibody against SARS-CoV-2 infection is also demonstrated in rhesus macaques. The authors suggest that fungus-produced antibody could overcome limitations with traditional mammalian protein expression systems, including reducing cost, reducing time, and increasing yields during antibody production. This could potentially be a useful system for producing monoclonal antibodies especially in the setting where rapid production is needed to keep up with emerging variants during a pandemic.

Major comments:

1. The paper uses a single antibody 87G7 against SARS-CoV-2 as proof of concept that the C1 expression system could be useful for production of human antibodies. It would have been helpful to see additional antibodies against SARS-CoV-2 (or other viruses) produced and characterized, at least in vitro, in this study.
2. Related to point 1, the discussion (starting at line 204) indicates that the authors have produced other antibodies, like nivolumab (data not shown). The yield of 87G7 appears to be approximately 14-fold lower than nivolumab. It would be helpful to know how the C1 expression system performs for other antibodies and if some antibodies are more difficult to produce, especially if this system is to be generalized as a platform technology.
3. One of the arguments for using this technology is that it may help with issues related to traditional mammalian expression systems, including “time required for development, variable yields, and high production costs”. How does the yield of 87G7 in the C1 expression system compare to the 293 transfection system. Further, comparison of yields to a CHO-expression system would be helpful since most clinical antibodies are produced with a CHO cells.
4. Monoclonal antibodies can also be produced by recombination in yeast. How does production in filamentous fungus compare to antibody produced in yeast with regards to yield, production time, and glycosylation? It would be helpful to have a discussion of advantages and disadvantages.
5. It would be helpful to mention the activity of 87G7 against contemporary, circulating variants, including XBB, BA.4/5, and BQ.1.1. This information is relevant since this is either a proof of concept study with an antibody that likely will not be further developed and/or a study supporting a pathway for further development of 87G7.

6. Figure 1: The non-reduced supernatant from the C1 expression system has significantly more lower molecular weight bands compared to the 293 expression system. These bands disappear when the samples are reduced and after purification (panel c). What do these bands represent? Are these other proteins secreted by C1 or are these degradation products of 87G7?

Minor comments:

1. The glycosylation pattern with the C1 expression system is interesting and potentially valuable for producing afucosylated antibodies that enhance NK cell ADCC activity. It would be helpful to also compare the binding in vitro of C1-produced vs 293-produced antibody to Fc gamma receptor IIIa.
2. The effect of antibody on viral replication/load is determined using TCID50 in hamsters and by qPCR in non-human primates. Was a viral titer determined in the non-human primate experiments? If so, it would be helpful to show that data since that could be a better reflection of live, replicating virus.
3. The methods section describes PRNT50 assays with sera. Is this method utilized in any of the results/figures?
4. Figure 3b and figure 4b appear to be showing the same assay (binding of C1- and 293-produced antibody to SARS-CoV-2 spike by ELISA), except with a different x-axis scale and curve-fitting.
5. Figure 5a: The filled bars make it challenging to see the individual data points.
6. Supplemental Figure 9: There is no mention of panel F in the legend.
7. Supplemental Table 2: The title states the data shows "virus isolation from NHP". However, this looks like data from immunostaining.

REVIEWER COMMENTS

Reviewer #1 (Remarks to the Author):

Kaiser et al have performed a comparison of monoclonal antibody production systems, mammalian versus a new fungal system, C1, using 87G7 as the Fab in IgG1 backbone against SARS-CoV-2 viruses. The experiments are clearly performed and well set out from in vitro characterisation to importantly hamster and non-human primate protection experiments. The lack of fucosylation of C1 produced antibodies is the main gain from the production method. The increase in Fc effector functions of the C1 and lack of fucosylation and shorter structures in the Fc as anti-inflammatory could be highlighted. This could also be extended to other viruses and diseases where Fc effector functions may be an advantage: Eg Rituximab fucosylation therapy is known to significantly impact tumor clearance (reviewed in <https://www.tandfonline.com/doi/abs/10.1517/14712598.6.11.1161>). Some clarity in the text would help for context and can be readily incorporated without further experiments needed.

We thank the reviewer for pointing out the review by Satoh et al. We agree that a more detailed discussion on the benefits of the afucosylated antibodies would further stress the gain of the C1 production method and improve the manuscript. We added some extra wording on the increase in Fc effector functions as a consequence lack of fucosylation, and the benefit of optimized Fc domains for prevention or treatment of COVID-19 and other conditions at line 274.

Other relevant advantages of the C1 expression system, related to time required for development, yields, and production costs, have been addressed as well in more detail in the discussion section (line 218, see below).

Minor comments:

- How was the Fab of the mAb derived? Where did it originally come from? At the initial results this could be given for context even if described in previous publications. This is buried at line 220, but should be described briefly in the introduction.

We appreciate the suggestion to briefly describe the origin of the monoclonal antibody 87G7 in the introduction to provide context. We have now included a brief statement in the introduction to address this at line 84: "In the current study, we have developed a C1 expression system for the well-characterized HuMab 87G7. This antibody was derived from H2L2 transgenic mice encoding the human immunoglobulin variable region immunized with the SARS-CoV-2 S protein. It binds to a patch of hydrophobic residues within the RBD of the SARS-CoV-2 Spike (S) protein and has broadly neutralizing activity against the VOCs Alpha, Beta, Gamma, Delta, and Omicron (BA.1/BA.2)."

- Which omicron variant used (BA.1 etc should be described) (line 535, BA.1 used, again buried in the text and could be briefly described earlier for context.

Although we mentioned the use of the specific omicron variant used in the methods section (line 535) as well as in the main text (line 149) "The therapeutic activity of the C1-derived antibody was evaluated by administration 12 h after infection with 10⁴ TCID₅₀ of Omicron BA.1 variant." we now added further reference at line 144 "The protective efficacy of C1-produced HuMab 87G7 was first assessed in a Syrian hamster challenge experiment using the SARS-CoV-2 Omicron BA.1 variant."

- Results line 106, Figure 1, what is the yield in comparison from the C1 vs HEK293 system? Input versus output, 1.6g/ml from C1 versus HEK293 system. If yield is the main advantage for reduced cost by C1 system, the yield or speed should be compared, a table would be helpful.

A direct comparison of yield between the C1 system and the transient HEK293 system or other expression systems is difficult to make specifically for this antibody. We have not optimized the different expression systems using the same construct. Ideally, a comparison would be needed with stably transduced CHO cells, which is the generally used strategies for mAb production. We now have added some more general discussion on this issue in the discussion section, line 218.

- Data not shown line 101, e.g. size integration, give the sequence (as per Nat Comms data availability?) Line 207, reference?

This data is considered basic cloning practice and was already mentioned in the methods section (line 327). We therefore deleted the sentence at line 101. The amplified fragment was confirmed by size assessment only and not by sequencing. Antibodies produced from these plasmids showed their functional integrity and were characterized in more detail.

We gathered more data regarding the expression of different antibodies in C1 and added some wording on this in the discussion (line 218). Given the fact that most of the data provided are part of a forthcoming publication we now referred to this information as unpublished observations.

- The data in Figure 4 is weakly described and vague
o Figure 4A, what is the source of the NK cells? PBMC ex vivo or cell line? Basis of activation? CD107a or IFN γ secretion? In the methods. The CD16+ NK cell source is not given.

We now have added the source of the NK cell-line NK92.05-CD16 (kind gift from Kerry S. Campbell) and the fact that cells were transferred to plates in the presence of a mouse monoclonal antibody to the human degranulation marker CD107a (V450-label, BD Biosciences) and golgistop and golgiplug (BD Biosciences), in the methods section (line 454) as well as the main text (line 140).

o What is the difference in Figure 4B and 3A, both appear to be S based ELISA assays, or is Figure 4B based on Fc γ R3a binding of the 2 mAb? The methods, line 435, described a cell-based ELISA? And flow cytometry, but no supporting FACS plots. This method is vaguely described, and the text and figure legend should be expanded.

Basically, Fig 4B and 3A both show indeed S based ELISA. Panel 4B is shown as an internal control of the same experiment to show that although binding is the same (Fig. 4B) NK activation is not (Fig 4A). Plates used for the NK activation were subsequently tested for S based ELISA (not a cell based ELISA). Flow cytometry was used to analyse the NK cell activation as mentioned in the methods section and main text (line 140).

- Supp figure 2 and Figure 3F, Area under the curve analysis for the VNA data maybe a useful way to summarise the 6 strains tested in the main figure as a panel.

We have calculated the IC50 of the different antibodies against the different variants and included a table with the results, now shown in Supplementary table 2.

IC50 (ng/ml)	614G	B.1.1.7	B.1.351	P.1	B.1.617.2	B.1.1.529
Humanized	7.864	7.771	3.74	12.54	4.895	5.29
C1	9.295	7.146	3.731	10.75	6.175	8.72

- Line 145/146, which figure panel does the 1 log reduction refer to?

Reference to the figure panel is now added at line 151 : “Viral titers were reduced by 1 log in the nasal turbinates, with no virus detected in the lungs of animals in which antibody had been administered 24 h before infection (Fig. 5A, B).

- Figure 6, Prophylaxis (-24 hours pi) and treatment (+12 hours pi) labels may help for context (or consistency w figure 5). What was the control treatment? An isotype or PBS control?

We now modified the labels of figure 6 for consistency with figure 5 and also replaced Control by isotype control to clarify this more clearly (figure 5). For consistency, Supplementary figure 3 was adapted as well.

- A HEK293 therapy group would have been a useful comparison, if not available due to difficulty in conducting hamster P3 experiments, a discussion point could be raised.

Indeed P3 logistics played a role in the execution of this experiment, but the therapeutic efficacy of this antibody had been demonstrated previously. We now added some discussion on this issue at line 158 “ Although we did not include a HEK293 therapy control group, earlier studies using the same antibody already showed a beneficial effect of the antibody when given after inoculation of the animals with the SARS-CoV-2 614G virus²¹.

- What antigen detection is Figure 5CD referring to? In figure legend but not main text, and which approach was used. Nucleocapsid detection by histology ELISA flow? This could be described briefly more explicitly.

Figure 5CD refers to nucleoprotein antigen detected using immunohistochemistry. This information has now been added (line 154) “...as determined by immunohistochemistry using an antibody against the nucleocapsid protein.”

- Supp figure 4 to 8, do not include treatment comparisons. Either time course of infection or treatment types would be informative, currently nothing is gained from these figures other than tissue context.

Indeed these figures show the tissue context depicting the lesions seen in the control animals. Quantitative assessments are shown in the figures 5, 6 and 8.

- Supp figure 4, the boxed areas could be directly labelled, rostral and caudal and nasal septum is unclear.

We now have modified Supplemental Figure 4 to include labelling of the rostral and caudal part. Labelling cross sections of the nasal septum is not possible in panel C.

- What is the treatment dose licensed for Palivizumab? 25mg/kg used. Hamsters are given mAb intraperitoneally, whilst macaque are intravenous, and slight difference in dose 20mg vs 25mg/kg, why the differences?

The treatment dose licensed for PalivizumabAb dosing is 15 mg/kg. The intravenous route of application was applied in macaques because of the high volume in which the antibody was provided for these studies, not allowing other routes. Similarly, dosing in the hamster experiment was adjusted at a later stage to 20 mg/kg in hamsters because of the maximum volume allowed to inject in relation to material available (mg/ml).

- Line 159, what are the statistical differences in lesions, mean stdev and p values- these stats could be described more throughout the main results text.

This information has now been added to the manuscript (line 162) : “ A significant reduction in histopathological lesions was observed in the respiratory ($P < 0.01$) and olfactory epithelium ($P < 0.01$) of the nasal turbinates, and in sections of whole lung lobes ($P < 0.05$) of animals treated therapeutically with C1-derived HuMab 87G7 (Fig. 6A-D and Supplementary Figs. 4-8). “

Reviewer #2 (Remarks to the Author):

The paper submitted by Kaiser et. al. characterizes a method for producing monoclonal antibodies by leveraging recombination and expression in the filamentous fungus *Thermothelomyces heterothallica* (C1 expression system). The SARS-CoV-2 monoclonal antibody 87G7 is used as proof of concept and antibody production, binding, glycosylation, viral neutralization, and efficacy in hamsters are compared between fungus-produced vs. 293 transient transfection-produced antibody. In vivo efficacy as prophylaxis and treatment of fungus-produced antibody against SARS-CoV-2 infection is also demonstrated in rhesus macaques. The authors suggest that fungus-produced antibody could overcome limitations with traditional mammalian protein expression systems, including reducing cost, reducing time, and increasing yields during antibody production. This could potentially be a useful system for producing monoclonal antibodies especially in the setting where rapid production is needed to keep up with emerging variants during a pandemic.

Major comments:

1. The paper uses a single antibody 87G7 against SARS-CoV-2 as proof of concept that the C1 expression system could be useful for production of human antibodies. It would have been helpful to see additional antibodies against SARS-CoV-2 (or other viruses) produced and characterized, at least in vitro, in this study.

We fully agree that further evidence beyond this proof of concept would be of interest. In fact several antibodies e.g. against other viruses as RVFV, ZIKA and Andes virus, have been produced in C1. However, these are early studies that need a detailed follow up and are not ready for publication. We now have added some wording on this at line 218.

2. Related to point 1, the discussion (starting at line 204) indicates that the authors have produced other antibodies, like nivolumab (data not shown). The yield of 87G7 appears to be approximately 14-fold lower than nivolumab. It would be helpful to know how the C1

expression system performs for other antibodies and if some antibodies are more difficult to produce, especially if this system is to be generalized as a platform technology.

Also in relation to this point raised we added some wording in the discussion (line 218) “ C1 has been developed to express and produce a variety of other proteins for therapeutic treatments, mainly mAbs but also proteins defined as “difficult to express” such as bi-specific and tri-specific antibodies and Fc-fusion proteins. Apart from Nivolumab we were able to express other mAbs (including antibodies against Rift Valley Fever virus and ZIKA virus) to high levels, ranging from 13.3 g/L to 24.5 g/L in 7-day fermentation (Tchelet et al., unpublished observations). “

3. One of the arguments for using this technology is that it may help with issues related to traditional mammalian expression systems, including “time required for development, variable yields, and high production costs”. How does the yield of 87G7 in the C1 expression system compare to the 293 transfection system. Further, comparison of yields to a CHO-expression system would be helpful since most clinical antibodies are produced with a CHO cells.

A direct comparison of yield between the C1 system and the transient HEK293 system or other expression systems is difficult to make. We have not optimized the different expression systems using the same construct. Ideally, a comparison would be needed with stably transduced CHO cells, which is the generally used strategies for mAb production.

We have added some further discussion on the use of the C1 expression system : “ However, a direct comparison of yield between the C1 system and the transient HEK293 system or other expression systems is difficult to make as we have not optimized the different expression systems using the same construct. Ideally, a comparison would be needed with stably transduced CHO cells, which is the generally used strategies for mAb production. C1 fermentation demonstrates significant benefits over CHO production as the C1 fermentation is based on fed-batch technology for 4-7 days with glucose feeding and with a defined low-cost medium and a wide range of conditions can be applied (pH: 5-8, Temp: 20°C - 45°C). In addition, the current strain doesn’t sporulate and the low viscosity culture allows relatively low power input compared to other fungal cultures. The protein production requires no inducer and the protein is typically secreted to the medium. The fermentation process can be easily scaled up (the largest fermentation volume with C1 so far was 500m3). Overall, the timeline for development of stable cell lines and production of purified protein through protein A purification is ~ 6 weeks after gene synthesis and the timeline from freezer to end of fermentation is much shorter. “

4. Monoclonal antibodies can also be produced by recombination in yeast. How does production in filamentous fungus compare to antibody produced in yeast with regards to yield, production time, and glycosylation? It would be helpful to have a discussion of advantages and disadvantages.

*Advantages and disadvantages have been discussed now at line 236: “ Yeasts on the other hand, including *S. cerevisiae*, *Pichia pastoris*, *Yarrowia lipolytica* and *Hansenula polymorpha*, are among the current heterologous expression platforms alternatives to CHO systems which are known to be robust, easy to genetically manipulate, cost-effective, and unlike *E. coli* possess native PTM machinery and lack endotoxins. However, in few cases, the expression level reached over 1-10 g/L . Under normal conditions, the protein production is obviously lower, especially with the expression of complex proteins. In addition, *P. pastoris* cannot produce or secrete all proteins to such titers. In comparison, C1 yields are generally higher, although this has not been studied head-to-head. C1 can use glucose*

and other cheap complex carbon source for fermentation while P. pastoris process has traditionally methanol and glycerol as the carbon sources. C1 can be fermented at variable conditions (pH: 5-8, Temp: 20°C - 45°C) in comparison to yeasts (typically 30.0°C and pH 5.0 to 6.0)."

5. It would be helpful to mention the activity of 87G7 against contemporary, circulating variants, including XBB, BA.4/5, and BQ.1.1. This information is relevant since this is either a proof of concept study with an antibody that likely will not be further developed and/or a study supporting a pathway for further development of 87G7.

Unfortunately, the activity of HuMab 87g7 against contemporary, circulating variants, including XBB1.5, BA.4/5, and BQ.1.1. is lost, primarily due to amino acid substitution at F486²¹. Now mentioned at line 263.

6. Figure 1: The non-reduced supernatant from the C1 expression system has significantly more lower molecular weight bands compared to the 293 expression system. These bands disappear when the samples are reduced and after purification (panel c). What do these bands represent? Are these other proteins secreted by C1 or are these degradation products of 87G7?

These bands are likely not due to proteolytic degradation of the mAb, because the reduced samples analysed in stained gel or in Western don't show any proteolytic fragments. The bands can be derived from heating the samples in a loading buffer (non-reduced samples without beta-mercaptoethanol) and running them in a gel with SDS. This may cause partial denaturation and disassembly of the antibody. It is noteworthy that also the HEK cell-produced 87g7 treated in the same way shows additional bands with lower molecular weight.

Minor comments:

1. The glycosylation pattern with the C1 expression system is interesting and potentially valuable for producing afucosylated antibodies that enhance NK cell ADCC activity. It would be helpful to also compare the binding in vitro of C1-produced vs 293-produced antibody to Fc gamma receptor IIIa.

A further validation of the C1 produced antibody and preferably others is foreseen, including further characterization of the interactions with different Fc receptors. Although we initially tried to perform these experiments in reply to the comment made by the reviewer, efforts failed thus far mainly due to logistic problems. It is anticipated that this will take too much time now. However, we agree with the main comment of reviewer 1 that specifically on this issue "Some clarity in the text would help for context and can be readily incorporated without further experiments needed." Discussion on this is added at line 274.

2. The effect of antibody on viral replication/load is determined using TCID50 in hamsters and by qPCR in non-human primates. Was a viral titer determined in the non-human primate experiments? If so, it would be helpful to show that data since that could be a better reflection of live, replicating virus.

Due to the low viral loads we did not titrate the non-human primate samples but rather tested qualitatively the presence of infectious virus in the sample (data shown in Supplementary Table 3).

3. The methods section describes PRNT50 assays with sera. Is this method utilized in any of the results/figures?

We thank the reviewer for this remark as this was indeed a mistake; antibodies were used instead of sera. We now adapted the text at line 552:

“ PRNT50 assays were performed as described previously. Briefly, monoclonal antibodies were 3-fold serially diluted in 60 μ L Opti-MEM I (1X) + GlutaMAX (Gibco). ”

4. Figure 3b and figure 4b appear to be showing the same assay (binding of C1- and 293-produced antibody to SARS-CoV-2 spike by ELISA), except with a different x-axis scale and curve-fitting.

Indeed the outcome of both assays is the same. However, the results of Figure 4b are shown to illustrate that in the same assay that was performed for the ADCC assay , equal binding to the spike proteins was observed as an internal control of the experiment. This has now been clarified more clearly in the legend of the figure : “ Binding of C1 and HEK293T produced HuMab 87G7 to the plate-immobilized ectodomain of SARS-CoV-2 S as measured by ELISA using the same plates that were used in panel A, to serve as an internal control to verify similar spike binding characteristics of the antibodies.” .

5. Figure 5a: The filled bars make it challenging to see the individual data points.

We now modified the figure to show the individual points more clearly.

6. Supplemental Figure 9: There is no mention of panel F in the legend.

We now have modified the legend and included a reference to panel F.

7. Supplemental Table 2: The title states the data shows “virus isolation from NHP”. However, this looks like data from immunostaining.

The data indeed show virus isolation from NHP. Immunostaining was done on the cells that became infected after isolation, to confirm that the induced cytopathic effect was due to SARS-CoV-2. This has now been clarified in the legend more clearly: “Calu-3 cells were inoculated with tissue homogenates and five days later cells were fixed. Cytopathic effects of cells were confirmed by immunostaining against SARS-CoV-2. + isolation positive; (+) unclear result; - isolation negative. ”

REVIEWERS' COMMENTS

Reviewer #1 (Remarks to the Author):

The authors have sufficiently addressed reviewer comments, and the manuscript is improved for clarity.

Figure 5 legend should refer to the support figures (4-8) of tissues they were quantified from as an example.

The different treatment routes and dosage used for the different animal models should be mentioned or discussed at the methods at the least.

Reviewer #2 (Remarks to the Author):

The authors have fully addressed the comments/concerns/feedback of this reviewer.

Reviewer #1 (Remarks to the Author):

The authors have sufficiently addressed reviewer comments, and the manuscript is improved for clarity.

Figure 5 legend should refer to the support figures (4-8) of tissues they were quantified from as an example.

We now have modified the legend of Figure 5.

The different treatment routes and dosage used for the different animal models should be mentioned or discussed at the methods at the least.

We now have mentioned the different treatment routes and dosages (if not already provided in the previous version) for the different animal models in the methods section of the manuscript.